# Erroneous predictive coding across brain hierarchies in a non-human primate model of autism spectrum disorder
Zenas C. Chao [1] ✉, Misako Komatsu [2,3,4] ✉, Madoka Matsumoto[5], Kazuki Iijima[5], Keiko Nakagaki[4] & Noritaka Ichinohe [4] ✉

In autism spectrum disorder (ASD), atypical sensory experiences are often associated with irregularities in predictive coding, which proposes that the brain creates hierarchical sensory models via a bidirectional process of predictions and prediction errors. However, it remains unclear how these irregularities manifest across different functional hierarchies in the brain. To address this, we study a marmoset model of ASD induced by valproic acid (VPA) treatment. We record high-density electrocorticography (ECoG) during an auditory task with two layers of temporal control, and applied a quantitative model to quantify the integrity of predictive coding across two distinct hierarchies. Our results demonstrate a persistent pattern of sensory hypersensitivity and unstable predictions across two brain hierarchies in VPA-treated animals, and reveal the associated spatio-spectro-temporal neural signatures. Despite the regular occurrence of imprecise predictions in VPA-treated animals, we observe diverse configurations of underestimation or overestimation of sensory regularities within the hierarchies. Our results demonstrate the coexistence of the two primary Bayesian accounts of ASD: overly-precise sensory observations and weak prior beliefs, and offer a potential multi-layered biomarker for ASD, which could enhance our understanding of its diverse symptoms.

Autism spectrum disorder (ASD) is a neurodevelopmental condition that includes challenges in social interaction and communication, repetitive behaviors, sensory hypo/hypersensitivity, and difficulties adapting to change. A leading mechanistic investigation of ASD focuses on its atypical sensory perception, such as hypersensitivities to light or sound, which is reported in around 90% of autistic adults[1]. Several theoretical models have been proposed to explain these sensory atypicalities. The enhanced perceptual functioning theory[2] and the weak central coherence theory[3] suggest that individuals with ASD have a bias toward locally-oriented processing, attending to details rather than global patterns. The temporal binding theory[4] suggests that individuals with ASD integrate sensory information over a prolonged time window, leading to a blurred or smeared perception of stimuli. The intense world theory[5] posits that excessive functioning of neural circuits causes heightened low-level sensory perception in ASD, leading to an overwhelming and fragmented sensory experience of the world. While these frameworks

significantly shape our understanding of ASD, they do not directly correspond to the underlying neural mechanisms.

In this study, we investigate ASD using the mechanistic framework of Bayesian inference[6,7], as its neural correlates are defined and accessible[8,9]. Through the Bayesian lens, sensory atypicalities in ASD could arise from various factors: overly precise sensory observations[10–12], weak prior beliefs[7,13], slow updates of these beliefs[14], and imbalanced control of precision[13,15,16], and overestimation of environmental volatility[17]. However, the corresponding behavioral evidence are inconsistent and conflicting. For example, prior beliefs in ASD have been shown to be both attenuated[18,19] and intact[11,20,21], and their variability has been reported to be both increased[22] and unaffected[11]. To directly test these Bayesian accounts, it is critical to identify their underlying neural implementations in ASD, which remains unknown.

The most promising implementation of Bayesian inference is predictive coding, which proposes that the brain creates internal models of the sensory world by a hierarchical and bidirectional cascade of large-scale

[1]International Research Center for Neurointelligence (WPI-IRCN), UTIAS, The University of Tokyo, 113-0033 Tokyo, Japan. [2]Institute of Innovative Research, Tokyo Institute of Technology, 226-8503 Tokyo, Japan. [3]RIKEN Center for Brain Science, 351-0198 Wako, Japan. [4]Department of Ultrastructural Research, National Institute of Neuroscience, National Center of Neurology and Psychiatry (NCNP), 187-8502 Tokyo, Japan. [5]Department of Preventive Intervention for Psychiatric Disorders, National Institute of Mental Health, National Center of Neurology and Psychiatry (NCNP), 187-8553 Tokyo, Japan. ✉e-mail: zenas.c.chao@gmail.com; mskkomatsu@gmail.com; nichinohe72@gmail.com

cortical signaling in order to minimize overall prediction errors[23–26]. Specifically, higher-level cortical areas predict inputs from lower-level areas through top-down connections, and prediction-error signals are generated to update the predictions through bottom-up connections when the predicted and actual sensory inputs differ. The theory has been applied to explain how atypical internal models are created in ASD[27,28]. Experimentally, prediction-error signals have been probed by surprise responses when expected stimuli are replaced or omitted. A key neural indicator of prediction error is the mismatch negativity (MMN), an event-related potential triggered by unexpected oddball stimuli, has been shown to vary in amplitude between individuals with ASD and typically developing individuals[29–31]. However, meta-analyses on these reports revealed no consistent trend in these differences[32,33]. Furthermore, the MMN amplitude can be influenced by statistical regularities over longer timescales[34,35], with this modulation found to be reduced in ASD[30]. This suggests that the interaction of prediction errors across hierarchical levels may be disrupted in ASD.

We hypothesize that the heterogeneous behavioral and neural evidence is caused by a diverse combination of erroneous predictive-coding computations occurring across cortical hierarchies, thus cannot be identified by a single neural representation where prediction-error signals across all hierarchies are mixed together. To test this hypothesis, we extract prediction-error signals across hierarchies and examine their atypical characteristics using a marmoset model of ASD[36]. This model was created by administering valproic acid (VPA) during pregnancy, a well-known risk factor for ASD. Maternal exposure to VPA induces ASD-like behavioral abnormalities and stress responses in marmoset offspring[37,38]. Importantly, the transcriptomic profile of the cerebral cortex in VPA-treated marmosets —reflecting the interactions between genetic and environmental factors— shows strong correlations with post-mortem brain transcriptomes from human ASD populations[36]. This correlation has not been observed in any rodent models previously used. Furthermore, the observed similarity in dysregulated neuronal gene networks between VPA-treated marmosets and humans with ASD suggests that this animal model could accurately represent major ASD subtypes, whose existence has been proposed due to weak interactions within individual gene networks[39,40].

To assess multi-level predictive coding, we use a local-global auditory oddball paradigm, where the subject passively listens to tone sequences with the temporal regularities established at two hierarchical levels[41]. This paradigm allowed a separation of hierarchical prediction-error signals[34,42–46]. To acquire large-scale neuronal dynamics with millisecond resolution, we use high-density hemisphere-wide electrocorticography (ECoG)[47]. To provide a mechanistic quantification of erroneous predictive coding, we use a hierarchical predictive-coding model that was previously used to disentangle prediction and prediction-error signals across hierarchies and quantify the integrity of prediction at each hierarchy[48].

Our results reveal sensory hypersensitivity and highly-variable predictions in the VPA-treated animals, which confirms the simultaneous presence of the two primary Bayesian accounts of ASD: overly-precise sensory observations and weak prior beliefs. Furthermore, we find distinct patterns of underestimation and/or overestimation of the sensory regularities at different hierarchies in the VPA-treated animals, supporting our hypothesis of erroneous hierarchical predictions as a source of ASD heterogeneity. Our findings map computational theories to their neural implementations and provide a potential neural marker for ASD that is multi-level, high-resolution, and mechanistic.

## Results

### Local-global auditory oddball paradigm to establish hierarchical regularities

Five marmosets, identified as Ji, Rc, Yo, Ca, and Rm, were used in this study. Among those, Ca and Rm were prenatally exposed to VPA (see Methods). During the task, subjects were seated with the head fixed and passively listened to a series of short tone sequences based on the local-global auditory oddball paradigm (Fig. 1a). Cortical activity was recorded with a 96-channel ECoG array covering nearly an entire cortical hemisphere (left hemisphere for Ji, Ca, and Rm, and right hemisphere for Rc and Yo) (Fig. 1b). For Rm, 5 channels in the orbital frontal area and 3 channels in the temporal area were surgical removed during the implantation due to tissue adhesions (88 channels remained).

During each trial, a series of 5 tones were delivered (Fig. 1a). The first 4 tones were identical, either low-pitched (tone A) or high-pitched (tone B) (jointly denoted as the standard tone x), and the fifth tone could be either the same (tone x) or different (jointly denoted as the deviant tone y). This resulted in two types of sequences: xx sequence (AAAAA or BBBBB) and xy sequence (AAAAB or BBBBA). Tone sequences were delivered in blocks of 100 trials, where two types of blocks were used: xx or xy blocks. In the xx block, 20 xx sequences were initially delivered as a standard sequence to habituate the subject; then there was a random mixture of 64 xx sequences (the trial type is denoted by xx|xx: xx sequence in xx block) randomly mixed with 16 xy sequences (xy|xx: xy sequence in xx block). Conversely, in the xy block, 20 xy sequences were initially delivered as a standard sequence, followed by a random mixture of 64 xy sequences (xy|xy: xy sequence in xy block) and 16 xx sequences (xx|xy: xx sequence in xy block).

This paradigm was designed to establish two levels of temporal regularity. A local regularity is established within a trial by the repetition of the first 4 tones, which is either followed or violated by the fifth tone. A global regularity is established by habituating the subject to a 5-tone sequence, which is either followed or violated by subsequent sequences. Local and global regularities are orthogonally varied, yielding four trials types: local and global standards (xx|xx), local and global deviants (xy|xx), local deviant but global standard (xy|xy), and local standard but global deviant (xx|xy).

**Fig. 1 | Local-global paradigm and ECoG layouts. a** The local-global paradigm and the tone and sequence designs. **b** The layout of the 96-channel ECoG arrays in the five subjects. For Rm, 8 channels were surgical removed after implantation due to tissue adhesion, and 88 channels remained.

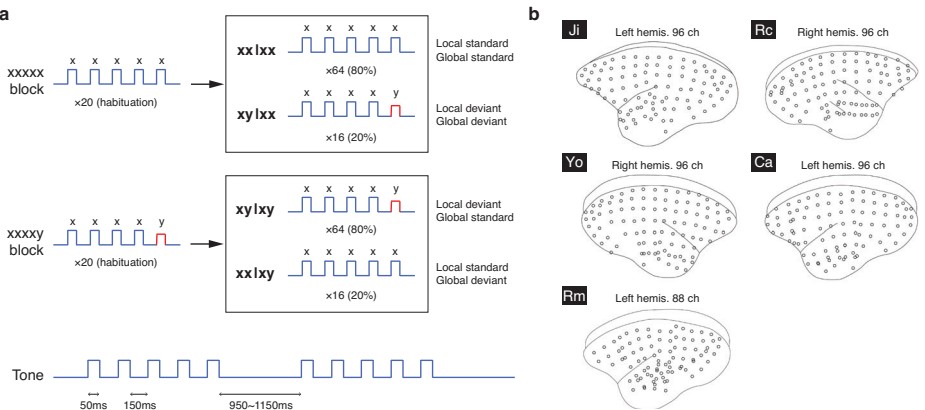

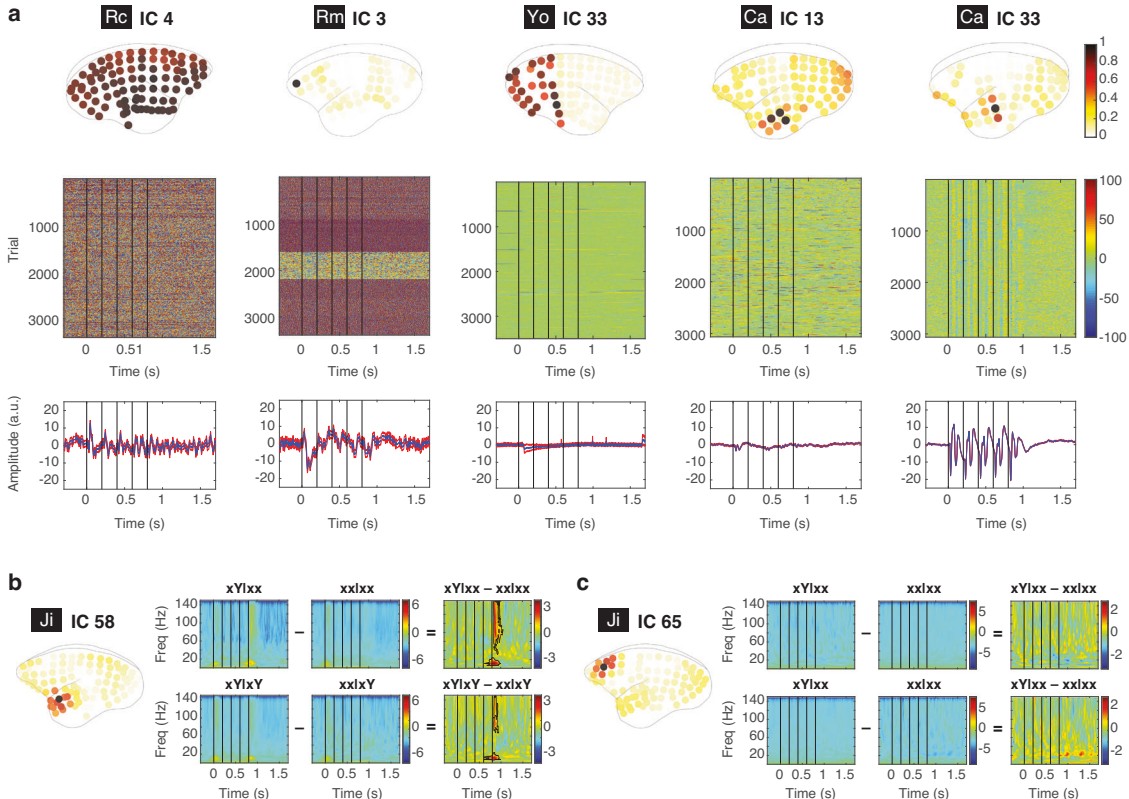

**Fig. 2 | Source signals and deviant responses. a** Examples of ICs. For each IC, the absolute spatial coefficients were normalized by the maximal value across electrodes, and shown on the top panel. The time courses of all the trials are shown in the second panel. Time zero represents the onset of the first tone, and the vertical black lines indicate the onsets of the 5 tones. The corresponding mean (blue) and standard error of the mean (red) across all trials are shown in the bottom panel. **b** The deviant responses from IC 58 in Ji. The spatial contribution of the IC is shown on the left. The ERSP for each trial type and the corresponding contrasts are shown. The black contours indicate the deviant response with a significant difference in ERSP in contrasts xy|xx – xx|xx and xy|xy – xx|xy. **c** Example of a non-significant IC. The same representation is used as in **b**.

## Deviant responses to local and global regularity violations

To examine the effect of VPA on how the local and global regularities were learned and represented in the brain, we evaluated the deviant responses in the brain when the regularities were violated. We compared ECoG signals from the xy and xx sequences in both the xx and xy blocks, i.e. xy|xx – xx|xx and xy|xy – xx|xy. By contrasting xy|xx and xx|xx trials, we can isolate deviant responses that arise when both local and global regularities are violated, i.e. a local deviant response that is also unpredicted by the global rule. Similarly, by contrasting xy|xy and xx|xy trials, we can capture the local deviant response that is predicted by the global rule.

To analyze the large-scale ECoG data, we first identified signal sources over the 96 electrodes (or 88 in Rm) by independent component analysis (ICA) (see Methods). Each independent component (IC) represented a cortical area with statistically-independent source signals (see examples of ICs in Fig. 2a). ICA could extract the reference signal (e.g. IC 4 in Rc, see Fig. 2a), artifacts introduced in different recording sessions (e.g. IC 3 in Rm), and artifacts introduced by the recording system (e.g. IC 33 in Yo). Moreover, ICA could help identify spatially-overlapped signal sources (e.g. ICs 13 and 33 in Ca). Therefore, our further analysis was performed based on individual ICs, instead of individual electrodes. See all ICs for each subject (96 for Ji, 92 for Rc, 88 for Yo, 90 for Ca, and 84 for Rm) in Fig. S1. Note that the numbers of ICs were different across subjects due to different numbers of bad channels were removed (see Methods).

The spatio-spectro-temporal dynamics of ECoG signals were quantified by the event-related spectral perturbation (ERSP) measured in decibel (dB) (with the baseline from 300 to 0 ms before the onset of the first tone, see more details in Methods). Each ERSP represents the in-trial cortical dynamics from an IC, during the time from 300 ms before the first tone to 900 ms after the fifth tone (a total of 600 time bins), across the frequencies between 0 and 150 Hz (a total of 150 frequency bins). Examples of ERSP for all four trial types and their contrasts are shown in Fig. 2b for IC 58 in Ji (located in the anterior temporal lobe) and in Fig. 2c for IC 65 in Ji (located in the dorsal prefrontal cortex). A deviant response was defined as a significant difference in ERSP, detected by a nonparametric cluster-based permutation test ($\alpha = 0.05$ corrected for multiple comparisons, two-sided, see Methods). An IC that showed deviant responses in xy|xx – xx|xx or xy|xy – xx|xy was identified as a *significant IC*. For example, IC 58 in Ji was a significant IC with deviant responses in both contrasts (Fig. 2b), while IC 65 in Ji was not (Fig. 2c). The numbers of significant ICs identified in Ji, Rc, Yo, Ca, and Rm were 5, 3, 4, 4, and 5, respectively. All the significant ICs are shown in Fig. 3. Also, see the deviant responses for all ICs in Fig. S1.

## Univariate analysis on deviant responses

To examine significant ICs, we first performed an univariate analysis to quantify their spatial, temporal, and spectral characteristics. To visualize the spatial distribution of each significant IC, its spatial coefficients were normalized to values between 0 and 1 by calculating their absolute values and then dividing them by the maximum. For each subject, the normalized spatial coefficients were then averaged across all significant ICs to obtain a joint topographic map (Fig. 4a). From the joint maps, we evaluated the relative contributions of three cortical areas: the posterior temporal cortex (pTC), the anterior temporal cortex (aTC), and the anterior prefrontal cortex (aPFC) (Fig. 4b). The brain areas were identified based on the Marmoset 3D brain atlas Brain/MINDS NA216[49] (see Methods). For each area, the relative contribution was quantified by the sum of the spatial

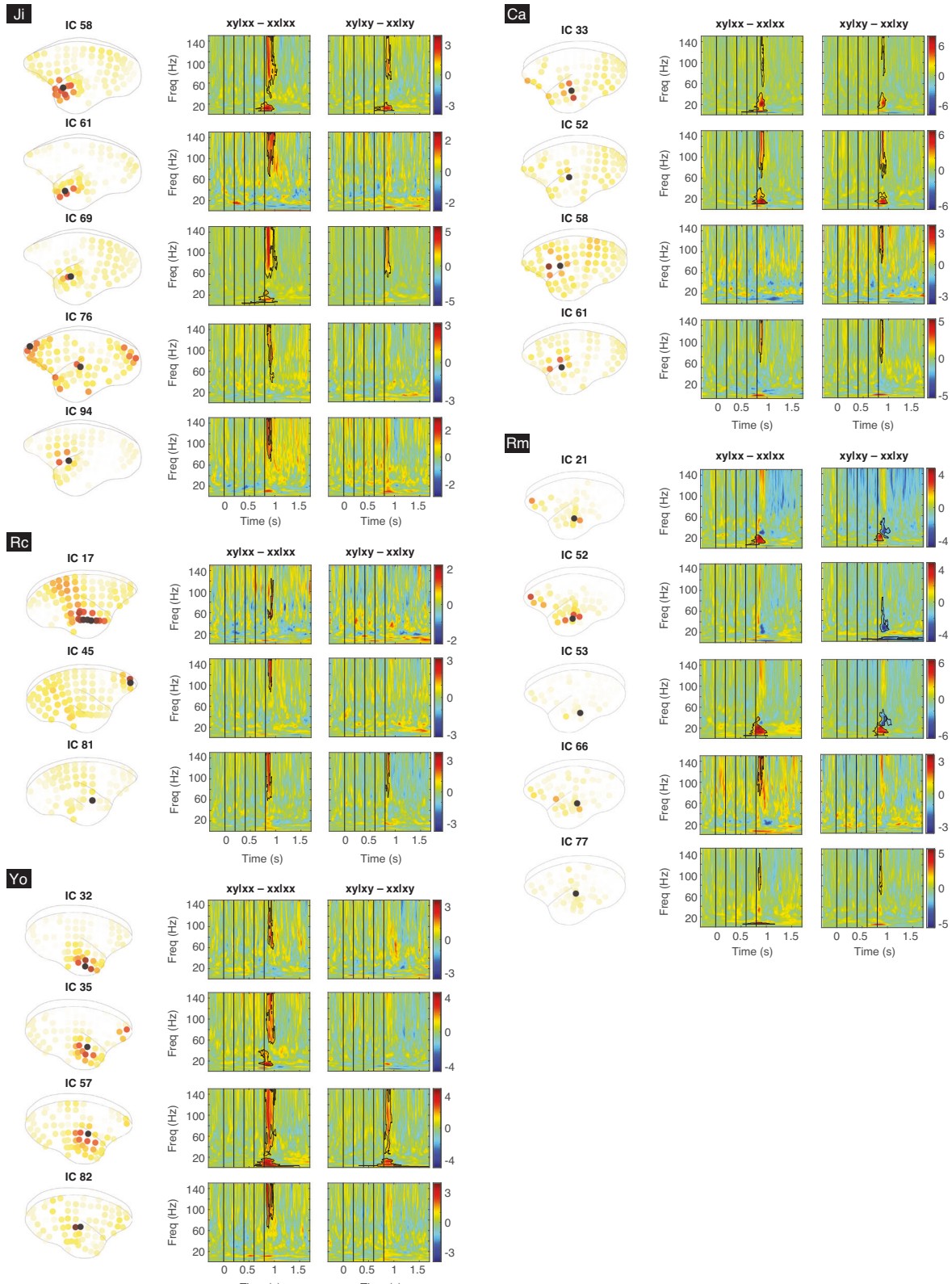

**Fig. 3 | Significant ICs from all subjects.** For each subject, all the significant ICs are labeled and shown with their absolute spatial contributions and the corresponding deviant responses in contrasts xy|xx – xx|xx and xy|xy – xx|xy (black contours).

distribution in the area divided by the total spatial distribution across all channels. For all subjects except Rm, the relative contributions from strong to weak were pTC > aTC > aPFC. For Rm, the contribution in aPFC was 22.0%, which was 2.6 times stronger the other subjects (8.6 ± 2.0%, $n = 4$ subjects).

To visualize the temporal and spectral distributions of significant ICs in each subject, absolute values of the deviant responses were averaged across all significant ICs to obtain a joint time-frequency representation (Fig. 4c). By averaging the joint deviant response across frequency bins, the peak responses after the last tone were found with comparable latencies of 67, 57,

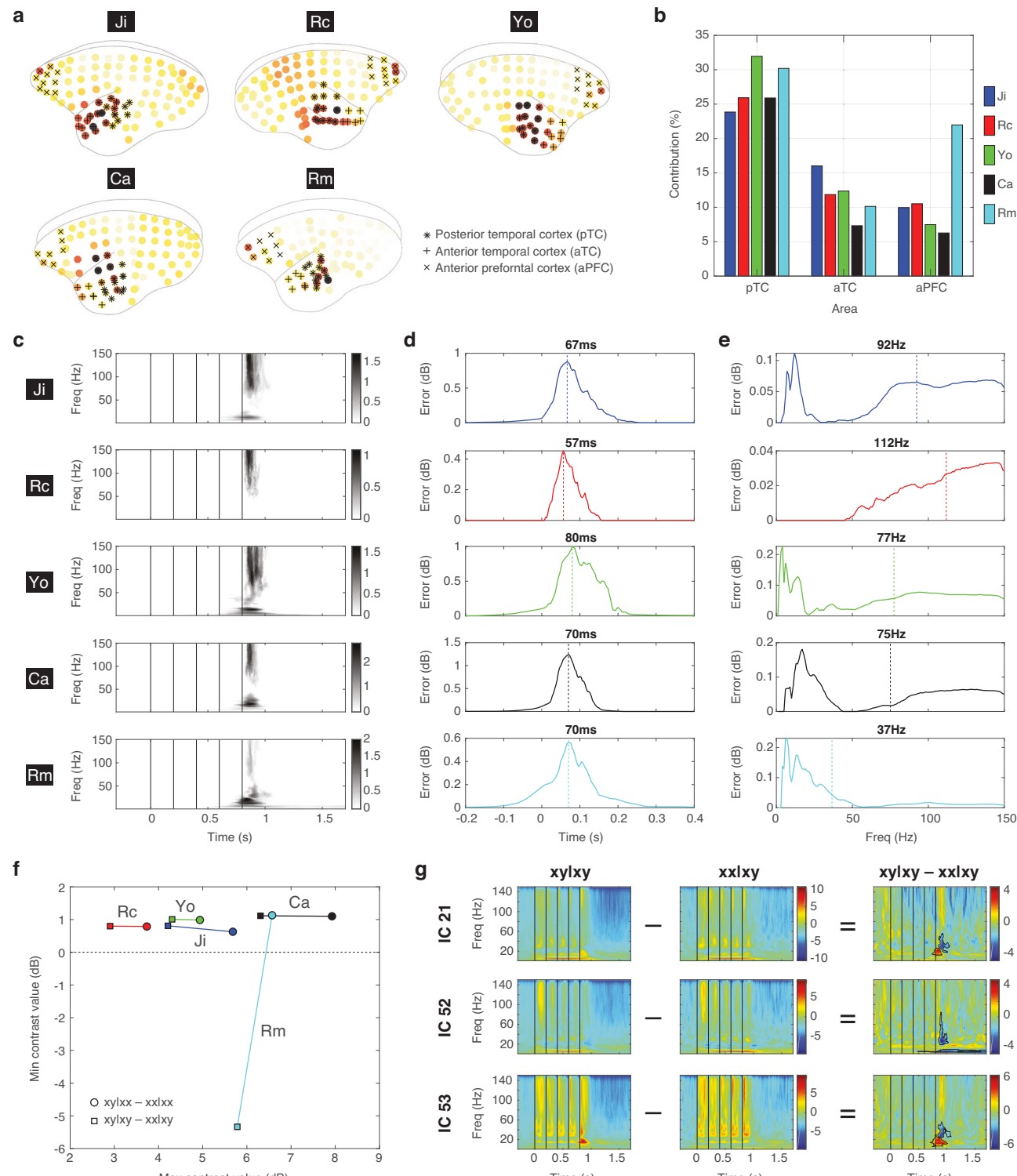

**Fig. 4 | Univariate analysis of the significant ICs. a** The joint topographic map of deviant responses for each subject. The electrodes on pTC, aTC, and aPFC are labeled with black star, plus, and cross signs, respectively. **b** The relative contributions of pTC, aTC, and aPFC for each subject. **c** The joint time-frequency representation of deviant responses for each subject. **d** The temporal profile of deviant responses. The peak response is indicated by a vertical dashed line and the latency is indicated. **e** The spectral profile of deviant responses. The average frequency is

indicated by a vertical dashed line and the value is indicated. **f** The maximal and minimal sizes of the deviant responses for each subject. The contrasts xy|xx – xx|xx and xy|xy – xx|xy are indicated by circles and squares, respectively. The color scheme is shared in panels **b**, **d**, **e**, and **f**. **g** The negative deviant responses in xy|xy – xx|xy in Rm for three significant ICs (21, 52, and 53, as in Fig. 3). The ERSP in xy|xy and xx|xy are also shown.

80, 70, and 80 ms for Ji, Rc, Yo, Ca, and Rm, respectively (Fig. 4d). By averaging the joint deviant response across time points, the average frequencies were found in the high gamma band at 92, 122, 77, and 75 Hz for Ji, Rc, Yo, and Ca, respectively, and in the low gamma band at 37 Hz for Rm (Fig. 4e).

We further evaluated the size of the deviant responses by measuring the maximal and minimal contrast values in the deviant responses across significant ICs (Fig. 4f). For all subjects, the maximal contrast values for contrast xy|xx – xx|xx were positive and greater than the maximal contrast values for xy|xy – xx|xy. This was consistent with the view that a greater surprise was evoked when both local and global regularities were violated (captured by xy|xx – xx|xx), while a smaller surprise was evoked when the local deviant was predicted by the global rule (captured by xy|xy – xx|xy). Furthermore, VPA-treated Ca and Rm showed stronger deviant responses than the unexposed Ji, Rc, and Yo. On the other hand, the minimal contrasts values were found to be positive, except in Rm where negative deviant responses were found in xy|xy – xx|xy. This is also shown in Fig. 3, where a power decrease in the beta/gamma bands (20 ~ 60 Hz) was observed in Rm for ICs 21, 52, and 53, particularly in xy|xy – xx|xy. To further examine the ERSP for those ICs in Rm, stronger responses to the last x tone in xx|xy were observed (Fig. 4g). This indicated a strong surprise toward the global deviant (last tone x in the xy sequence), and suggested that Rm was more sensitive to the violation of the global rule.

The univariate analysis revealed some abnormal characteristics in the deviant responses in Ca and Rm. In summary: (1) VPA-treated Ca and Rm showed stronger deviant responses than the unexposed, suggesting their hypersensitivity to deviant stimuli; (2) hyperactivity in the prefrontal cortex was found in Rm, not Ca, which could link to its hypersensitivity to the global regularity; (3) high-gamma deviant responses, which were thought to represent bottom-up prediction errors, were absent in Rm.

## A hierarchical predictive coding model for the local-global paradigm

To further investigate how sensory sensitivity and erroneous predictions could lead to the observed abnormal deviant responses, we used a model-fitting analysis based on a quantitative model of hierarchical predictive coding[42,48]. The quantitative model we used has previously been shown to effectively explain the brain responses during the local-global paradigm with a goodness-of-fit closed to the optimal data-driven decomposition, allowing for mechanistic evaluations of sensory sensitivity and prediction strengths at both local and global levels[48].

The model describes the interactions between prediction and prediction-error signals during the last tone of a sequence after both local and global regularities are learned (see Fig. 5a). It consists of three hierarchical levels (Level S, Level 1, and Level 2) and two streams (x stream and y stream). Level S is the sensory level that receives thalamic input, which was a value between 0 and 1, Level 1 learns and encodes the local regularity, which is the tone-to-tone transition probability (TP), and Level 2 learns and encodes the global regularity, which is the sequence probability (SP). The x and y streams process the tone x and y, respectively.

The predictive coding operations across hierarchies in two streams are illustrated in Fig. 5a. In the x stream, Level S receives a sensory input (assumed to have a strength of 1) and a prediction signal ($P1_x$) from Level 1, and sends a prediction-error signal ($PE1_x$) to back Level 1. Moving up, Level 1 receives the prediction-error signal from Level S and a prediction signal from Level 2 ($P2_x$), and sends a prediction signal to Level S and a prediction-error signal ($PE2_x$) to Level 2. Lastly, Level 2 receives the prediction-error signal from Level 1, and sends a prediction signal to Level 1. In a manner comparable to the x stream, the y stream also features prediction signals $P1_y$ and $P2_y$, along with prediction-error signals $PE1_y$ and $PE2_y$.

Based on the model, the strengths of the prediction signals ($P1_x$, $P2_x$, $P1_y$, and $P2_y$) are to minimize the mean-squared error received at that level, and can be determined once the transition and sequence probabilities are known. Once the strengths of the prediction signals are determined, the local prediction error (PE1 = $PE1_x + PE1_y$) and the global prediction error (PE2 = $PE2_x + PE2_y$) in the deviant responses evoked by the last tone can be calculated by subtracting the model values for xy|xx – xx|xx and xy|xy – xx|xy. For an in-depth understanding of the model and its calculation, see the Supplementary Information and the corresponding Figure S2.

## Models with erroneous sensory sensitivity and hierarchical predictions

In the model, the local and global prediction errors are obtained under the optimal predictions, where the mean-squared prediction errors are minimized at each level. To further evaluate the potential erroneous sensory sensitivity and hierarchical predictions for the VPA-exposed, we further added some tunings to the model across different levels (Fig. 5b).

At Level S, a scaling factor $s_0$ was added to the sensory input in the x stream to account for the sensory sensitivity or adaptation for the repetitive tone x (left panel). The value of $s_0$ was between 0 and 1, where $s_0 = 1$ represents no sensory adaptation or no diminished responses to repeated exposure of tones. For the xy sequence, since tone y does not repeat, adaption does not occur in the y stream (right panel). At Levels 1 and 2, we added scaling factors $s_1$ and $s_2$ to the first-level predictions ($P1_x$ and $P1_y$) and the second-level predictions ($P2_x$ and $P2_y$), respectively, to account for imperfect predictions. When $s_1 = 1$ and $s_2 = 1$, the predictions are optimal. When $s_1 < 1$ or $s_2 < 1$, the prediction underreacts to the input (sensory input or first-level prediction error, respectively), i.e. "hypo-prediction", and is insufficient to cancel it out. For example, if $s_1 = 0$, there will be no first level prediction, and the prediction errors continue to propagate to Level 2 without reducing. When $s_1 > 1$ or $s_2 > 1$, the prediction overreacts to the input, i.e. "hyper-prediction", where the corresponding transition or sequence probabilities are overestimated and additional errors are created. Note that $s_1$ and $s_2$ were applied to both the x and y streams, since erroneous estimation of transition or sequence probabilities could occur at both streams.

## Model-fitting for optimal decomposition of deviant responses

Now we have a predictive coding model tailored to the local-global paradigm, characterized by only three parameters: $s_0$, $s_1$, and $s_2$. This model was then utilized to determine which parameter combination most accurately accounts for the deviant responses observed in ECoG data. To achieve this, we first pooled all deviant responses (as shown in Fig. 3) to create a tensor with three dimensions: *Contrast*, *IC*, and *Time-Frequency* for the functional, anatomical, dynamical aspects of the data, respectively. For each subject, the dimensionality of the tensor was 2 (xy|xx – xx|xx and xy|xy – xx|xy) by 3 ~ 5 (the number of significant ICs) by 90,000 (600 time points and 150 frequency bins).

We then factorized the 3D tensor into PE1 and PE2 components by performing parallel factor analysis (PARAFAC)[50], setting the first dimension according to the model-derived values (see Methods). This model-fitting analysis was performed for 9261 ( = 21 × 21 × 21) models, each with a unique combination of the scaling factors $s_0$ (21 values between 0 and 1), $s_1$ (21 values between 0 and 2), and $s_2$ (21 values between 0 and 2). For each model, the goodness-of-fit was evaluated by the residual sum of squares (RSS) and core consistency[51]. The best-fitting model was determined as the one with the smallest RSS and a core consistency above 80%.

The parameters of the best-fitting models for all subjects are shown in Fig. 5c. The best-fitting models for the unexposed animals (Ji, Rc, and Yo) were found with similar scaling factors: $s_0 = 0.35 ~ 0.45$, $s_1 = 0.8 ~ 0.9$, and $s_2 = 0.7 ~ 0.8$. For Ca, the best-fitting model was found when $s_0 = 0.75$, $s_1 = 0.3$, and $s_2 = 0.2$. This suggested a hyper sensory sensitivity ($s_0$ was twice the size as for the unexposed) and hypo-predictions at both the local and global levels. On the other hand, the best-fitting model for Rm was found when $s_0 = 0.95$, $s_1 = 1.0$, and $s_2 = 1.72$. This indicated that Rm shared a similar hyper sensory sensitivity as in Ca, but with a normal local prediction and a hyper global prediction.

In summary, the model-fitting analysis revealed potential mechanisms that cannot be observed by univariate analysis, and indicated that (1)

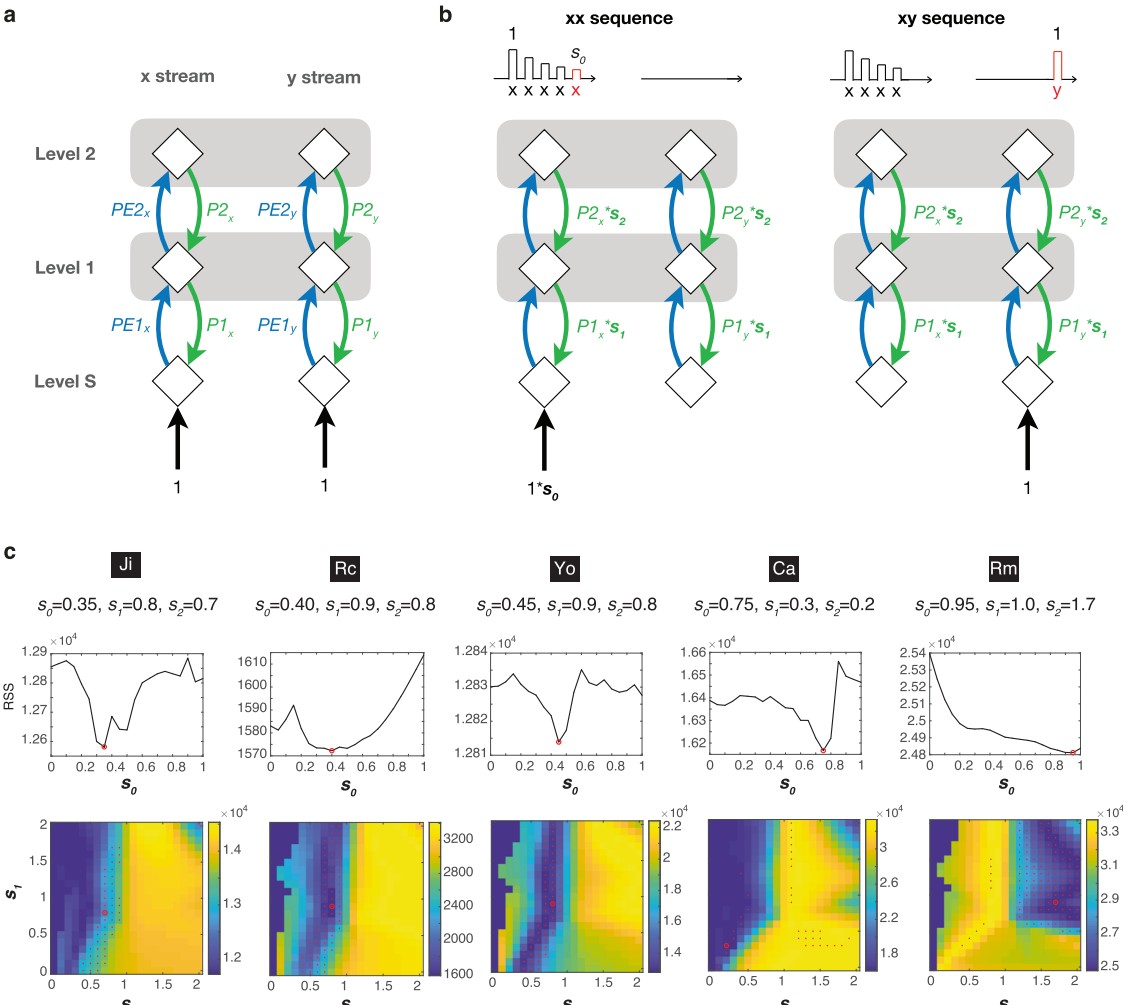

**Fig. 5 | A quantitative predictive coding model and the optimal parameters for model-fitting. a** The proposed neural operations for predictive coding across streams and hierarchies. **b** Model tunings with $s_0$, $s_1$, and $s_2$. A decreased response (scaled by $s_0$) to repeated tone x during the xx sequence (left), and a fresh response to tone y during the xy sequence (right) (the last tones are shown in red). The corresponding models are also shown, where the sensory input to the x stream is scaled by $s_0$, and P1 and P2 are scaled by $s_1$, and $s_2$, respectively. **c** The model-fitting results for each subject are shown in each column, where the optimal parameters are indicated. For each $s_0$, the minimal RSS across different combinations $s_1$ and $s_2$ is shown in the top panel. The minimal RSS is indicated by a red circle. The combination of $s_1$ and $s_2$ under this minimum is indicated by a red circle in the bottom panel. Models with a fitting consistency >80% are indicated by red dots. The color bar represents RSS.

predictions in the unexposed animals were close to optimal at both hierarchical levels, (2) hyper sensory sensitivity was found in both VPA-treated animals, and (3) different types of erroneous hierarchical predictions were observed between VPA-treated animals.

**Prediction-error signals extracted from best-fitting models**
Next we visualized the spatial, spectral, and temporal signal patterns of the PE1 and PE2 components extracted from the best-fitting models. These components were visualized by their composition in the three tensor dimensions. The first dimension showed how much PE1 and PE2 contributed to the deviant responses in the two contrasts (Fig. 6a), which was determined by the model and used for the model-fitting. The model values were different across subjects, since different optimal parameters were obtained.

The second dimension showed the contribution of each significant IC to PE1 and PE2 (Fig. 6b). For example, in Ji, only ICs 58, 69, and 94 contributed to PE1, and IC 69 contributed the most. To further visualize these contributions on a brain map, the normalized spatial coefficients of significant ICs (as in Fig. 3) were combined based on their contributions (see more details in Methods). The resulting brain maps are shown in Fig. 6b. Overlaps between PE1 and PE2 were observed for most subjects, but primarily PE1 appeared in the posterior temporal cortex and PE2 appeared in

the anterior temporal cortex and the anterior prefrontal cortex. This propagation of prediction errors from the temporal cortex to the prefrontal cortex is consistent with previous evidence from both monkey and human studies using the local-global paradigm or its variations[34,41–44].

The third dimension showed the in-trial spectro-temporal dynamics for PE1 and PE2 (Fig. 6c). To examine the temporal dynamics of PE1 and PE2, we averaged the time-frequency representation in Fig. 6c across all frequency bins (Fig. 6d). PE1 peaked at 47, 53, 80, 67, and 37ms after the last tone, while PE2 peaked later at 93, 93, 133, 100, and 103ms for Ji, Rc, Yo, Ca, and Rm, respectively. To examine the spectral profiles of the PE1 and PE2 components, we measured their maximal activation at each frequency bin across all time bins (Fig. 6e). The average frequencies were 97, 112, 81, 95, and 30 Hz for PE1, and 100, 121, 90, 64, and 41 Hz for PE2 in Ji, Rc, Yo, Ca, and Rm, respectively. The high-gamma components were absent in Rm, as described in the univariate analysis (Fig. 4e).

**Response variability underlying deviant responses**
For Rm, the absence of high-gamma components in the deviant responses (as in Fig. 6e) could result from two possibilities: (1) the sizes of prediction-error signals carried in the high-gamma band were comparable between the xx and xy sequences, or (2) the sizes of prediction-error signals were different between the xx and xy sequences but the trial-to-trial variability was

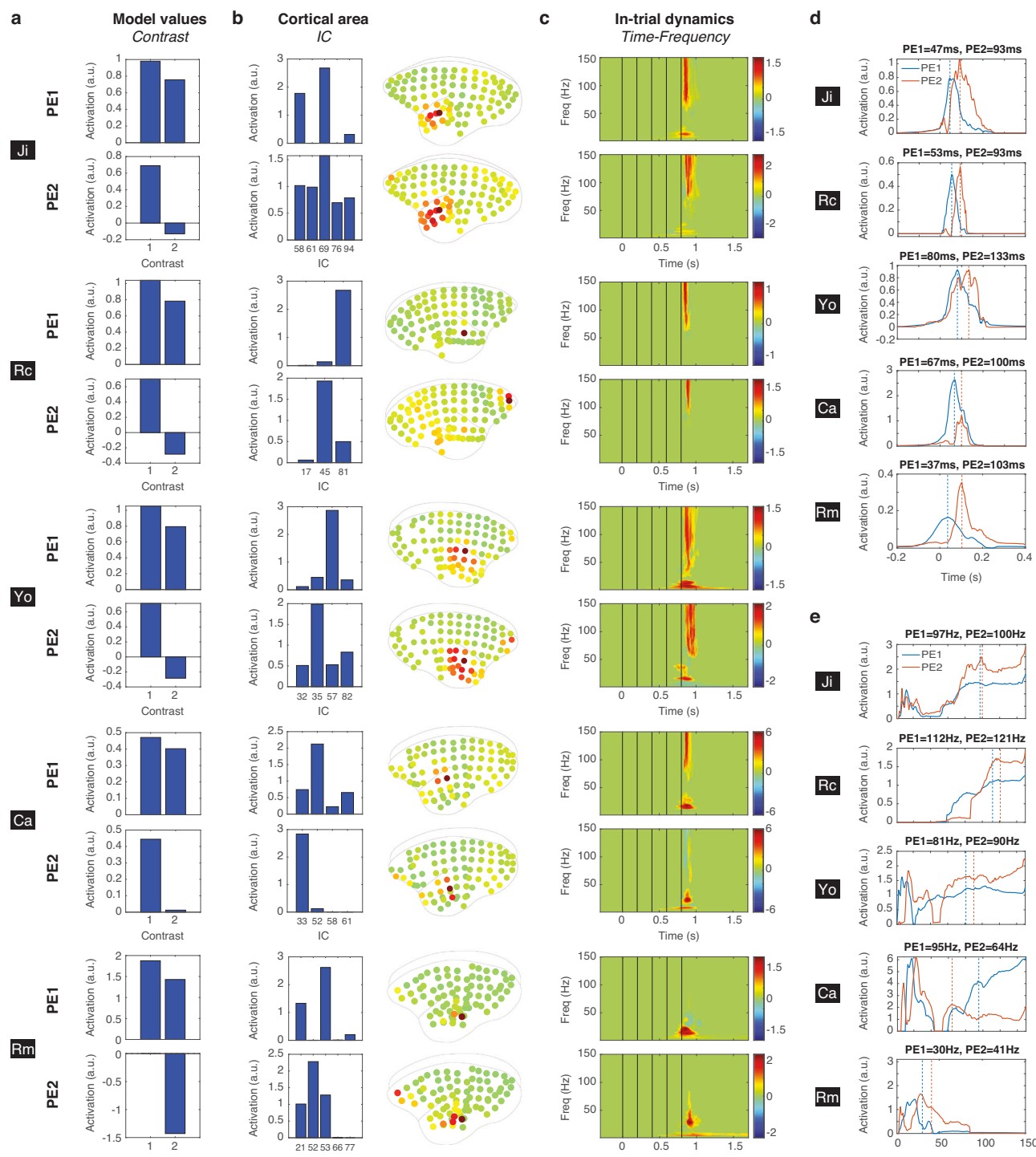

**Fig. 6 | Prediction-error signals extracted from the best-fitting model. a** The contributions of PE1 and PE2 to the deviant responses in the two contrasts, which are based on the model values from the best-fitting model. **b** The spatial dimension of the PE1 and PE2 components extracted from the best-fitting model. The contribution of each significant IC to PE1 and PE2 and the corresponding average brain maps are shown. **c** The spectro-temporal dimension of the PE1 and PE2 components extracted from the best-fitting model. **d** The temporal profiles of PE1 (blue) and PE2 (orange). The maximal activations are indicated as vertical dashed lines and the corresponding peak latencies are shown. **e** The spectral profiles of PE1 and PE2. The average frequencies activations are indicated as vertical dashed lines and the corresponding values are shown.

too high to obtain statistical significance. The former suggests that no prediction was established, and the latter suggests that the prediction was highly variable over trials.

To test these two possibilities, we quantified the sizes of PE1 and PE2 on a trial-by-trial basis by projecting the single-trial EEG responses onto the spatio-spectro-temporal structures of PE1 and PE2. First, we obtained the

spectro-temporal structures averaged across subjects, with a focus on the high-gamma band (indicated by red contours in Fig. 7a). Then, we mapped the ERSP of each significant IC to these averaged structures, assigning weights based on their contributions in the model-fitting process (as in Fig. 6b) (see Methods for comprehensive details). As results, two projection values were obtained for each trial, quantifying the presence of PE1 and PE2

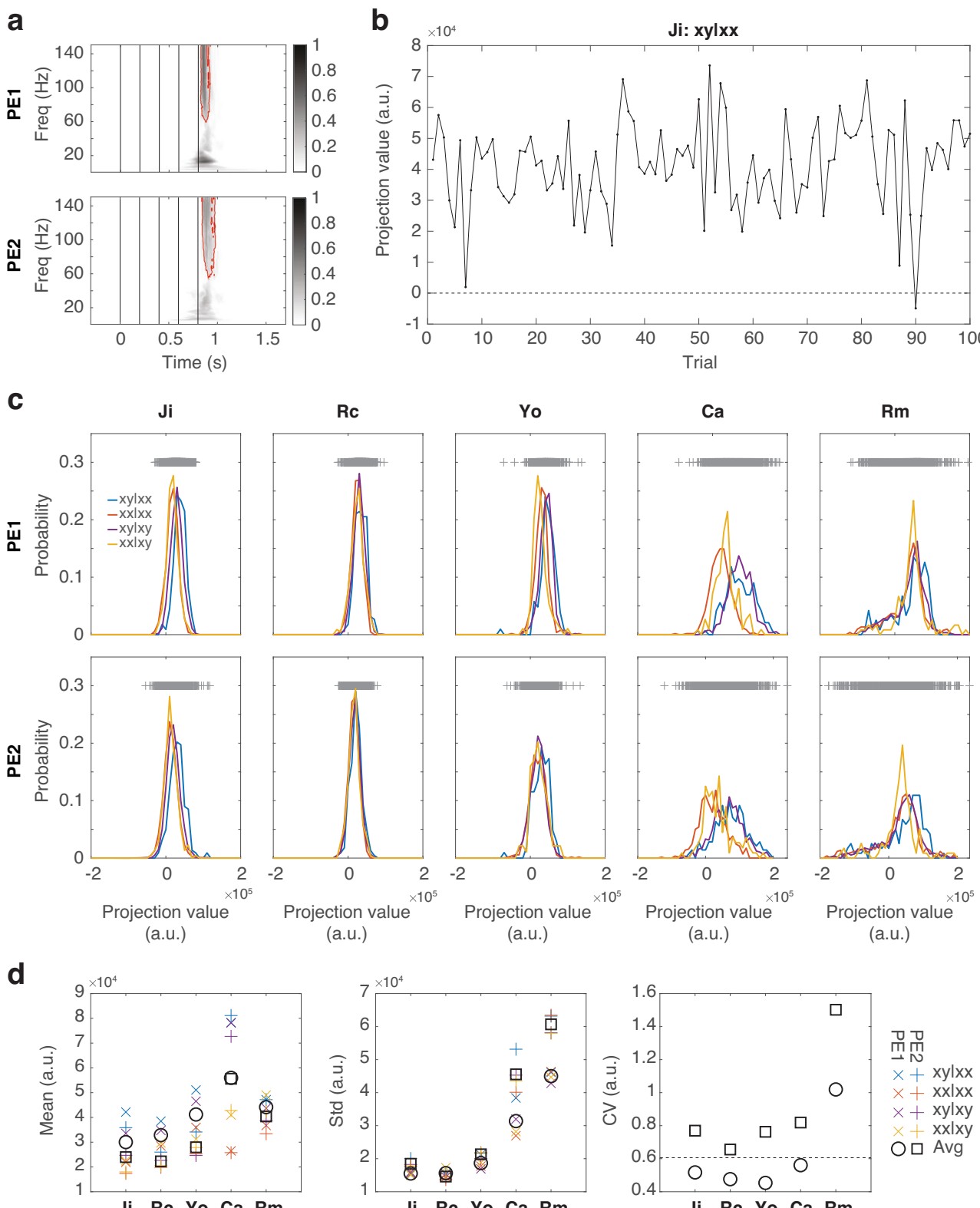

**Fig. 7 | High signal variability in VPA-treated animals. a** The spectro-temporal structures of PE1 and PE2. The masks of the top 75% values in the high-gamma band (>40 Hz) are indicated by red contours. **b** An example of the projection values during 100 trials of xy|xx in Ji. **c** The probability distributions of the projection values for PE1 (top row) and PE2 (bottom row) in each subject. The probability distribution for each trial types are shown in different colors. Projection values across all trials are indicated as gray crosses on the top. **d** The mean, standard deviation, and CV of the projection values. The mean and standard deviation measured for each trial type are labeled with different colors. The average values across all trial types are indicated by black circles and squares for PE1 and PE2, respectively. CV calculated from the average values are shown. The horizontal dashed line indicates the mean CV from the unexposed animals (across PE1 and PE2).

in the high-gamma responses. An example of the projection values over trials is shown in Fig. 7b.

We quantified the probability distributions of the projection values for PE1 and PE2 for each trial type and subject (Fig. 7c). Wider distributions were found in the VPA-treated animals, suggesting that the predictions and their subsequent prediction errors were highly-variable at both the local and global levels. We further quantified the mean, standard deviation, and coefficient of variance (CV, the standard deviation divided by the mean) of the projection values (Fig. 7d). The mean values for both PE1 and PE2 were higher in Ca and Rm when compared to the unexposed animals, with Ca exhibiting higher mean values than Rm (Wilcoxon rank sum test, two-sided, $\alpha = 0.05$). The high mean values in Ca were consistent with the findings that Ca had high sensory sensitivity and the subsequent high prediction errors were not adequately explained away due to hypo-predictions.

The standard deviations for both PE1 and PE2 were higher in Ca and Rm when compared to the unexposed animals (Wilcoxon rank sum test, two-sided, $\alpha = 0.05$). In Rm, the high standard deviation with the comparable mean value led to high CV, which supported the second possibility that the absence of the high-gamma components was resulted from highly-variable predictions. In Ca, the high standard deviation was compensated by the high mean value, which led to low CV and the significant high-gamma components.

### Sensitivity analysis on model parameters

The results shown in Figs. 5, 6, and 7 were obtained from the optimal models selected based on the criterion of achieving a fitting consistency higher than 80%. Employing different criteria may result in a varied set of candidate models, which in turn could identify alternative optimal models. Considering that a consistency between 80 to 90% is indicative of a robust decomposition, and a consistency above 40% is viewed as somewhat satisfactory[52,53], we conducted a sensitivity analysis on the consistency threshold. In this analysis, we varied the consistency threshold from 40% to 100% in 1% increments to determine the optimal parameters ($s_0$, $s_1$, and $s_2$), resulting in a total of 61 optimal models. For each optimal model, we also measured the signal variability (SV), which was the mean standard deviation in the projection values (as in Fig. 7d).

For all optimal models, we plot SV against $s_0$ (Fig. 8a) and $s_1$ against $s_2$ (Fig. 8b). For both SV and $s_0$, the unexposed group (Ji, Rc, and Yo) showed significantly lower values compared to the VPA-treated group (Ca and Rm) (Wilcoxon rank sum test, two-sided, $\alpha = 0.05$) (see Fig. 8c). For $s_1$ and $s_2$, no significant difference was found between the unexposed and the VPA-treated groups (p-value = 0.0743 and 1 for $s_1$ and $s_2$, respectively). This suggested that the VPA-treated group exhibited unstable predictions and heightened sensory sensitivity, without consistent patterns of erroneous local and global predictions relative to the unexposed group.

We conducted further comparisons between individual subjects (see Fig. 8d). No significant differences were found among the unexposed animals for SV, $s_0$, $s_1$ and $s_2$ (Wilcoxon signed rank test, paired and two-sided, $\alpha = 0.05$, Bonferroni multiple-comparison correction). Compared to the unexposed animals, Ca showed significantly lower $s_1$ and $s_2$ values, whereas Rm showed significantly higher $s_2$ values. This indicated that the VPA-treated animals exhibited individual differences in their erroneous local and global predictions.

## Discussion

We combine a passive auditory paradigm with a quantitative model to extract the neural signatures of hierarchical prediction-error signals, and evaluate the integrity of predictive coding in VPA-treated animals. Through this approach, we unveil both sensory hypersensitivity and unstable predictions in VPA-treated animals. Notably, these fluctuating predictions present distinct patterns of underestimation and/or overestimation of hierarchical sensory regularities, potentially contributing to the diverse characteristics of ASD. By linking computational theories with their neural underpinnings, our study contributes to the foundation for potentially identifying a comprehensive, multi-tiered, and mechanistic neural marker for ASD.

The hierarchical organization of prediction-error signals in the auditory local-global paradigm has been examined in humans and non-human primates[34,43–46,54]. Utilizing both data-driven and model-driven analyses to decompose predictive-coding signals that are not only interdependent but also spatially and temporally overlapping, we have identified the neural signatures of local and global prediction-error signals in macaque ECoG[42] and human EEG[48]. The spatio-spectro-temporal markers observed in both macaques and humans bear similarity to those depicted in Fig. 6, showing prediction-error signals in high frequency band ( >30 Hz) propagating from the auditory cortex to the frontal cortex with a delay of ~50 m. This suggests a shared neural organization that facilitates hierarchical predictive coding in both human and non-human primates. Moreover, it indicates the potential applicability of our animal model findings to human patients.

The local-global paradigm has been utilized to study atypical perception and emotion processing in ASD. In a study of adults with ASD, a smaller MMN was found in the ASD group than in the typically developing (TD) group[30]. Moreover, both groups demonstrated a reduced MMN when the global rule could be anticipated, such as in the xy block, but this reduction was more pronounced in the TD group than in the ASD group. This implies weaker local and global predictions among individuals with ASD. In a study with children (8 ~ 15 year old) with ASD, no significant differences in MMN were found between ASD and TD groups, suggesting that local prediction error was processed normally[29]. When manipulating the global rule to establish various levels of expectation for local deviants, children with ASD responded differently. There was a decrease in fronto-cortical responses to sequences that were unexpected, whereas there was an increase in late frontal activation in response to anticipated sequences. These findings suggest that there may be abnormalities in global prediction within the ASD population. In addition, individuals with ASD demonstrated MMN in response to violations of local emotion regularity for both faces and music, but their responses to global emotion regularity violations were absent[55]. These results, derived from a group level analysis, suggest a potential deficiency in global prediction within ASD. They are in alignment with other discoveries of unusual contextual modulations in sensory processing in ASD, observed in both non-social settings[56,57] and social contexts[58–60]. However, our results demonstrate the importance of conducting an individual-level analysis, which would help in identifying the potential diversity in abnormalities in multi-level regularity processing.

Our findings indicate that the VPA-treated animals exhibited elevated $s_0$ values, suggesting that their responses to repetitive stimuli were not as significantly reduced compared to the healthy controls. We interpret this lack of sensory adaption as heightened sensory sensitivity, and use this evidence to support the overly-precise sensory observations account of ASD[10–12]. However, sensory hypersensitivity can also be attributed to imbalanced precision controls, where the brain faces challenges in prioritizing sensory information based on its perceived reliability or precision[13,15,16]. In this case, there is a tendency to assign higher weight to low-level sensory details, potentially intensifying the sensitivity to sensory stimuli. Importantly, these theories are not mutually exclusive and can provide complementary insights into the understanding of sensory processing in ASD. To gain a deeper understanding of their respective contributions, employing a trial-by-trial analysis with Bayesian modeling that incorporates precision parameters could be beneficial[61,62]. Additionally, conducting experiments that effectively control the precision of stimuli can provide valuable insights into the interplay between sensory processing and precision weighting[63].

It is worth noting that our model does not differentiate the origins of this presumed sensory adaption to repetitive stimuli, only its outcome. One possible cause is stimulus-specific adaptation, an inhibitory neuronal mechanism observed in both cortical and subcortical structures[64–66]. Another possible cause is predictive coding itself, where the prediction of transitions between identical tones is learned during repetitions, and the repetitive tones generate less surprise over time. To fully explain the data will require a model that includes the interplay between and stimulus-specific

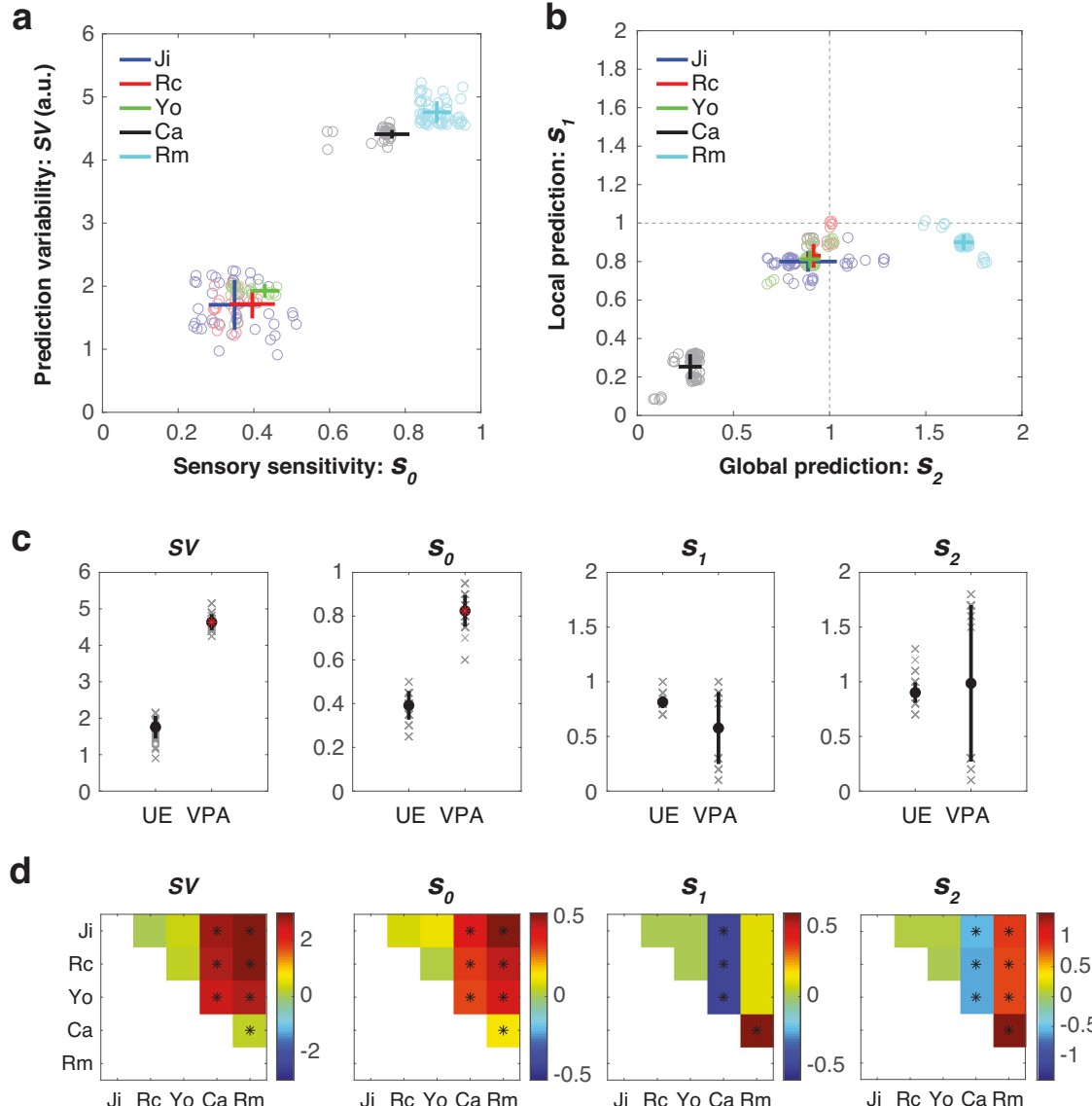

**Fig. 8 | Sensitivity analysis on model parameters. a** The signal variability ($SV$) vs. $s_0$. values from each optimal model are represented by circles, with the mean and standard deviation illustrated by a bar. When numerous values cluster closely together, slight random variations are introduced to spread them out, making them easier to distinguish visually. The color scheme corresponds to that used in Fig. 4b. **b** $s_1$ vs. $s_2$. The same presentation is used as in **a**. **c** Comparisons between the unexposed (UE) and VPA-treated groups. For each variable, the individual data and the corresponding mean and standard deviation are shown, and the significant differences are denoted by red asterisks. **d** Comparisons between individual subjects. For each variable, we illustrate the differences by subtracting the values of the subject on the y-axis from those on the x-axis. Significant differences are indicated with asterisks.

adaptation and predictive coding to describe the neural dynamics during each tone in both cortical and subcortical areas.

Our findings demonstrate erroneous predictions across different cortical hierarchies in VPA-treated animals. This implies a deficiency in perceptual learning, which is a key characteristics of the ASD phenotype[67]. This deficiency could stem from abnormalities in synaptic plasticity[68] or learning-related changes in neural connections[69], and lead to slow belief updates[14]. Beyond irregular predictions, another consideration is the potential misestimation of the volatility or uncertainty associated with these predictions. This "belief in volatility" is thought to play a crucial role in driving learning[70], and in modulating the balance between exploiting existing knowledge and exploring new possibilities[71]. Studies have shown that adults with ASD tend to overestimate the volatility of sensory environments, leading to a greater tendency to ignore unexpected stimuli and experience less surprise[17]. However, it is still uncertain whether underestimations of volatility can occur in ASD, possibly leading to contrasting

patterns of prediction errors. To explore this, further manipulation of environmental uncertainty and a detailed examination of the learning processes are required.

Regrettably, our current model has limitations in assessing the learning process as it only represents the signals once the temporal regularities have already been learned and the errors have been minimized. To understand the dynamic process of prediction updating and error minimization, it is crucial to investigate the trial-by-trial signaling that occurs during the learning process. A Bayesian model known as the hierarchical Gaussian filtering (HGF) has emerged as a promising candidate for understanding prediction updating and error minimization[72]. This model employs precision-weighted prediction errors[23,25,73] and has been utilized to investigate prediction-error signals in the brain during learning[74–80]. HGF models the stimulus probability (first level), the tendency of the change of the probability over a longer timescale (second level), the volatility of the tendency (third level), where higher levels determine the rate of change at lower

levels. It's important to note that this hierarchical representation differs from predictive coding, where higher levels influence the state of lower levels. Therefore, the sequence structure in the local-global paradigm is not simply a tendency of stimulus probability or its volatility. Nevertheless, exploring whether the higher levels of HGF can capture sequence probability will be a worthwhile endeavor. Another candidate is dynamic causal modeling (DCM), which is also a Bayesian model that can be used to estimate the coupling among brain regions and the changes in coupling over time and across experimental conditions[81]. This method has been utilized in studying MMN[61,82,83], albeit with the constraint that the brain regions of interest had to be predetermined.

The phylogenetic closeness between nonhuman primates and humans, particularly in molecular, circuitry, and morphological features of the brain, has made these species increasingly attractive as novel models of psychiatric disorders. Boasting a well-differentiated frontal lobe and intricately stratified hierarchical cortical connectivity, their cerebral cortex closely resembles that found in humans[84]. Marmosets, in particular, offer many advantages as model organisms for the developmental disorder autism. For example, early sexual maturation, high reproductive rates, efficient space utilization due to their compact size, the potential for genetic manipulation, and a complex repertoire of social skills[85].

The VPA-treated model marmosets utilized in this study show deficits in social tasks that require sophisticated and hierarchical internal models[86,87]. This includes the ability to adjust one's motivation based on observations of others' behavior and to evaluate the reciprocity of others. It is important to note that gene expression within the marmoset cortex closely matches the postmortem brains of human ASD, suggesting that it more closely resembles humans than any prior rodent model[36]. Notably, gene expression associated with myelin and inhibitory neurons, which are thought to be important for brain computation, is commonly reduced in both individuals with ASD and the VPA-treated marmosets. Continued studies of hierarchical predictive coding using the VPA-treated marmosets may provide important insights into the nature of human ASD. This is especially important given the profound social deficits exhibited in ASD.

A major limitation of our study is the small number of VPA-treated marmosets, which was primarily due to the complexities involved in administering VPA and implanting hemisphere-wide ECoG. This restricts our ability to assess the diversity and spectrum of erroneous predictive coding and to identify potential ASD subtypes. Additionally, as the diagnosis of ASD can be influenced by sex[88], and sex differences are anticipated in both neurotypical and atypical brain development[89], the fact that our study groups comprised only male animals precludes an investigation into whether sex contributes to the observed variability in erroneous predictive coding. Moreover, in our study, the unexposed group underwent both right and left implantations, while the VPA-treated group only had left implantations. Including more subjects with implantations on each side will allow us to investigate potential atypical lateralization in ASD[90] and reduce biases associated with functional lateralization[91].

Another limitation of our study is the absence of behavioral measurements, which restricts our understanding of VPA's effects. To address this, our future plans include integrating our current experimental and analytical approaches with multidimensional behavioral indices, endocrine and autonomic nervous system data, and brain transcriptome data. This integration aims to enhance our capacity to draw parallels between model findings and the complex pathology of ASD in humans. Another future plan is to examine the single-trial learning using a more generalized HGF that can bridge more directly to predictive coding[92].

In summary, we record large-scale high-resolution neural data in a non-human primate model of ASD and identify different neural signatures underlying different predictive coding accounts of ASD. This research has the potential to contribute to the identification of neural markers specific to different subtypes of ASD and shed light on the impact of prenatal VPA exposure on neurodevelopmental pathways leading to ASD.

## Methods

### Animals
We used five adult common marmosets (Callithrix jacchus; all males, 320–450 g, 22–42 months). Before the ECoG arrays were implanted into the monkeys, they were familiarized with the experimenter and experimental settings. The animals had ad libitum access to food and water throughout the experimental period. Two animals (Ji and Rc) were raised and recorded at RIKEN Center for Brain Science, and the other three animals (Yo, Ca, and Rm) were raised and recorded at the National Center of Neurology and Psychiatry (NCNP). Marmosets were housed in an environment maintained on a 12/12 h light/dark cycle, and given food (CMS-1, CLEA Japan) and water ad libitum. Temperature was maintained at 27–30 °C and humidity at 40–50%.

All procedures of the ECoG study at RIKEN were conducted in accordance with a protocol approved by the RIKEN Ethical Committee. All procedures of the VPA preparation and ECoG study at NCNP were conducted in accordance with NIH guidelines and the "Guide for the Care and Use of Primate Laboratory Animals" published by the National Institute of Neurological Research, National Center of Neurology and Psychiatry, and approved by the Animal Research Committee of NCNP.

**VPA treatment.** The method used to produce VPA-treated marmosets followed the same procedure outlined in previously published work[36]. In short, serum progesterone levels in the female marmosets were monitored once a week to determine the timing of pregnancy. In addition to the blood progesterone level, pregnancy was further confirmed by palpitations and ultrasound monitoring (Ultrasound Scanning; Xario, Toshiba Medical Systems Corp., Tochigi, Japan). We orally administered 200 mg/kg of sodium valproate (VPA, Sigma–Aldrich, St. Louis, MO, USA) seven times from day 60 to 66 after conception to the mother marmosets. We did not observe obvious malformations or deformities in VPA-treated marmosets.

### Electrode implants
The whole-hemisphere 96-channel ECoG arrays (Cir-Tech Co. Ltd., Japan) were chronically implanted. We epidurally implanted the array into the right hemisphere for Rc and Yo, and the left hemisphere for Ji, Rm and Ca. Eight electrodes (channels 92 ~ 94) from Rm were cut during the implantation due to tissue adhesions. The surgical procedures for electrode implantation have been previously described in detail[47]. The coordinates of recording electrodes were identified on the basis of the combination of pre-acquired MR images and postoperative computer tomography images using AFNI software[93] (http://afni.nimh.nih.gov). Then, we estimated the location of each electrode on cortical areas by registering to the Marmoset 3D brain atlas Brain/MINDS NA216[49] with AFNI and ANTS[94].

### Experimental setup
ECoG signals from monkeys Ji and Rc were acquired at RIKEN using a Grapevine NIP system (Ripple Neuro, Salt Lake City, UT) at a sampling rate of 1 kHz. Experiments of monkeys Yo, Rm, and Ca were conducted at NCNP. The neural signals were stored at a 1017.25 Hz sampling resolution into a TDT signal processing system RZ2 (Tucker-Davis Technologies, Alachua, FL). During the ECoG recordings, the marmoset was seated in a primate chair in an electrically shielded and sound-attenuated chamber with their head fixed. The auditory stimuli were delivered bilaterally by two audio speakers (Fostex, Japan) at a distance of ~ 20 cm from the head at an average intensity of 65 dB SPL.

### Stimuli and experimental procedure
Two tones with different pitches (Tone A = 800 Hz; Tone B = 1600 Hz) were synthesized. Each tone was 50 ms in duration. Series of five tones were presented with a 150 ms inter-tone interval, with 950–1150 ms was set between the offset of the last tone of a sequence and the onset of the first tone of the following sequence (see Fig. 1a). Four different stimulus blocks were used: AAAAA, BBBBB, AAAAB, and BBBBA blocks. In AAAAA blocks, 20

AAAAA sequences were delivered, followed by a random mixture of 64 AAAAA and 16 AAAAB. In BBBBB blocks, 20 BBBBB sequences were delivered, followed by a random mixture of 64 BBBBB and 16 BBBBA. In AAAAB blocks, 20 AAAAB sequences were delivered, followed by a random mixture of 64 AAAAB and 16 AAAAA. In BBBBA blocks, 20 BBBBA sequences were delivered, followed by a random mixture of 64 BBBBA and 16 BBBBB. In each experimental day, we conducted ECoG recordings on 1 ~ 8 blocks, depending on the animal's condition. For each animal, we performed 7-9 recordings for each block.

## Data analysis

**Preprocessing and independent component analysis (ICA).** The ECoG signals were downsampled to 300 Hz by EEGLAB on MATLAB[95] (function: pop_resample.m). Bad channels were then removed by visual inspections: channels 6, 7, 8, and 80 were removed in Rc; channels 8, 68, 71, 81, 83, 85, 87, and 88 were removed from Yo; channels 2, 44, 48, 61, 63, 64 were removed in Ca, channels 1, 6, 7, 8, 9, 10, 43, 43, 49, and 81 were removed in Rm. For each subject, all data were concatenated together, and ICA was performed by the FieldTrip Toolbox[96] (function: ft_componentanalysis.m with the runica algorithm). For each trial, the ICA signals were aligned at the onset of the first tone, and signals from 0.3 s before to 1.7 s after the onset of the first tone were segmented and used for the further analyses.

**Event-related spectral perturbation (ERSP).** For each subject, independent component (IC), and trial, the time–frequency representation of the ICA signal was generated by Morlet wavelet transformation at 150 different center frequencies (1 ~ 150 Hz) with the half-length of the Morlet analyzing wavelet set at the coarsest scale of 7 samples, which is implemented in the FieldTrip Toolbox (ft_freqanalysis.m). Baseline normalization was then performed to calculate the decibel values by using the baseline period from –0.3 to 0 s (time zero as the onset of the first tone) (ft_freqbaseline.m).

**Deviant response.** For each subject, the deviant responses (xy|xx – xx|xx and xy|xy – xx|xy) were calculated for each IC across all trials. To measure the significance of the difference in ERSP (as the black contours shown in Fig. 2b), we performed permutations by shuffling trial indices, and used a nonparametric cluster-based method for multiple comparisons correction[97], which is implemented in FieldTrip Toolbox (ft_freqstatistics.m with 500 permutations). Non-significant values in the deviant responses were set to 0, and ICs with no significant deviant responses in both xy|xx – xx|xx and xy|xy – xx|xy were considered as nonsignificant ICs.

**Model-fitting with parallel factor analysis (PARAFAC).** We used PARAFAC, a generalization of principal component analysis (PCA) to higher-order arrays[50], which was previous used for the computational extraction of latent structures in functional network dynamics[42,98]. To decompose deviant responses into components with theorized contrast values, PARAFAC was performed with the first dimension *Contrast* fixed with the values proposed by the model. This was done by the N-way toolbox[99], with no constraint on all three dimensions (using FixMode and OldLoad inputs in parafac.m). The convergence criterion (i.e., the relative change in fit for which the algorithm stops) was set to $1e-6$. The initialization method was set to be direct trilinear decomposition (DTLD), which was considered the most accurate method[100]. For each fitting, the residual sum of squares (RSS) and the core consistency diagnostic[51] were measured.

**Brain spatial contribution.** For each significant IC, the absolute values of the spatial filter (1 × number of channels) were first calculated and normalized by their maximal value (as in Fig. 3). The brain map shown in Fig. 6b was the linear combination of the normalized spatial filters of all significant ICs and their contributions in the model-fitting (Fig. 6b).

**Single-trial projection.** The spectro-temporal structures of PE1 and PE2 were determined by normalizing the corresponding time-frequency representations (as in Fig. 6c) to values between 0 and 1 and then averaged across subjects. A mask in the high-gamma band was determined as the top 75% values in frequencies above 40 Hz (red contour in Fig. 7a). The single-trial contributions of PE1 and PE2 were then obtained by projecting single-trial EEG responses of each significant IC onto these masks.

For each subject, a single-trial ERSP response (ERSP= number significant ICs × 150 frequency bins × 600 time points) was projected on a PE1 or PE2 spectro-temporal mask (FT = 600 time points × 150 frequency bins) and the contributions of all significant ICs (S = 1 × number significant ICs) (as in Fig. 6b). This was calculated as S*ERSP*FT, which yields a single scalar value. Note that the spectro-temporal masks for PE1 and PE2 were calculated from all subjects and thus shared across subjects, while the contributions of all significant ICs were different across subjects.

## Model calculation

The detailed model calculation is described in Supplemental Information, and the MATLAB code for these calculations is provided.

## Statistics and reproducibility

The number of trials used is comparable to previous similar studies[34,42,44,54], and the raw ECoG data is provided. The model is fully described through equations, and the MATLAB code required for its calculations is included. In terms of data analysis, we detail the variable dimensionality, MATLAB toolboxes, functions, and key parameters used. For statistical comparisons, we include details on the number of resamplings and the methods used for multiple comparisons. The only subjective aspect of our methodology is the ECoG preprocessing, where bad channels were manually excluded based on visual inspection. We adhered to a general guideline, resulting in an exclusion of 6 ± 4% (mean ± standard deviation) of electrodes across different sessions (0%, 4%, 8%, 5%, and 11% for Ji, Rc, Yo, Ca, and Rm, respectively).

## Reporting summary

Further information on research design is available in the Nature Portfolio Reporting Summary linked to this article.

## Data availability

Source data underlying main figures are presented in Supplementary Data 1. The raw ECoG data can be freely downloaded (https://dataportal.brainminds.jp/ecog-auditory-02).

## Code availability

The code to calculate values of predictions and prediction errors in the proposed model has been deposited in Zenodo[101].

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

## Acknowledgements

We thank Yuri Shinomoto and Takaaki Kaneko for animal care and awake recordings; Junichi Hata for obtaining the MRI images. We also thank Dr. Wataru Suzuki for his assistance in setting up the ECoG lab system and marmoset experiments and Ms. Akiko Tsuchiya for her technical support to marmoset breeding and the creation of the VPA-treated marmoset. This work was supported by World Premier International Research Center Initiative (WPI), MEXT, Japan (to Z.C.C.), Brain/MINDS from the Japan Agency for Medical Research and Development (JP20dm0207001 and JP20dm0207069 to M.K.; JP23dm0207066 to N.I.), JSPS KAKENHI (JP 23H04978 to M.K.), JST Moonshot R&D (JPMJMS2294 to M.M), and an Intramural Research Grant (Nos. 23-7, 26-9, and 29-6 to N.I.; 5-8 to M.K.) for Neurological and Psychiatric Disorders from the NCNP.

## Author contributions

Z.C.C. conceptualized the study. M.K. and Z.C.C. refined the experimental protocol. M.K. coordinated and conducted ECoG experiments. K.I. and M.M. conducted ECoG experiments at NCNP. N.I. and K.N. provided VPA marmosets, including marmoset rearing. Z.C.C. designed and performed the data analysis. Z.C.C. wrote the first draft of the paper, and N.I. and M.K. helped with the editing. All authors contributed to and have approved the final paper.

## Competing interests

The authors declare no competing interests.
