## [Peer Review File · Communications Biology]

Reviewers' comments:

Reviewer #1 (Remarks to the Author):

Article revision

Title

Erroneous Predictive Coding Across Brain Hierarchies in a Non-Human Primate Model of Autism Spectrum Disorder

Overview

Atypical sensory experience in Autism Spectrum Disorder (ASD) is associated with irregular predictive processing across brain hierarchies. This study employed high-density electrocorticography to record neural activity during passive listening to a local-global oddball paradigm. The subjects were typical developing and valproic acid-induced female and male marmosets.

The results reveal the diverse configuration of prior beliefs, demonstrating the coexistence of overly precise and weak priors within the valproic acid-induced group, and a common hypersensitivity.

These findings support the existence of heterogeneous sensory profiles within the ASD population.

Major comments

Introduction

- Page 3, lines 2-6: The introductory paragraph is quite lax and somewhat weak. More importantly, it is not informative on the environmental valproic acid-induced animals used in this study. We recommend addressing the validity and robustness of autistic traits shown by this model, and its wide use in basic research.

- Page 3, line 10: Sensory atypicality in ASD is explained by different theories such as “intense world theory”, “temporal binding hypothesis”, “atypical hierarchical information processing” (Balasco et al., 2019). It should be clarified that failures in Bayesian inference are one theory among others attempting to explain atypical sensory experiences in ASD.

- Page 3 and 4, lines 25 and 1-6, respectively: “Experimentally, prediction-error signals have been probed by surprise responses when expected stimuli are replaced or omitted” is a lax introduction to surprise responses. We recommend rephrasing to emphasize that mismatch negativity (MMN), the brain response to the violation of the regularity, is considered the neural substrate of prediction error (Fitzgerald and Todd, 2020). The lines 1-6 are confusing. The MMN is not one example, but the brain response to the surprise event interrupting a regular context. This response is recorded by EEG, MEG, and fMRI, as some examples (Garrido et al., 2009). The authors must rephrase this paragraph for the sake of clarity.

- Page 4, lines 8-11: Single- or multiple neural representations of predictive coding computations are provided by the complexity of the oddball paradigm (i.e., classical oddball paradigm vs. local-global oddball paradigm, respectively). The MMN is elicited by violating the regularity of either oddball paradigm. Indeed, the MMN frames the experimental design chosen for this study on local-global processing. The authors should reformulate this paragraph.

Methods

- Page 24, line 4: This study used female and male marmosets to address autistic traits without specifying biological sex of typical developing and valproic acid-induced animals. In the induced

group, the diverse findings on prior beliefs may be influenced by sex, as sex differences are expected in the typical and atypical brain (Ruigrok et al., 2014). Moreover, ASD's diagnosis is biased by sex (Gu et al., 2023). Thus, it is crucial to specify whether sex is a factor of variability when analyzing neural responses in the valproic acid-induced group.

- Page 25, line 8: The array was implanted into the right or left hemisphere of the marmosets. The typical developing group had 2 right and 1 left implantations, and the valproic acid-induced group had 2 left implantations. Apparently, hemisphere location does not influence neural recordings. The authors should record a single hemisphere per experimental design to avoid future biases (i.e., Devlin et al., 2003). Describe as a limitation to avoid in future studies.

Minor comments

Introduction

- Page 4, line 8-9: Correct the verb in the following sentence: "We hypothesize that the heterogeneous behavioral and neural evidence is caused by a diverse combination of erroneous predictive-coding computations ""occurring"" across cortical hierarchies [...]"

Reviewer #2 (Remarks to the Author):

I want to state first that my field of expertise is limited to theoretical and empirical discussions around predictive coding and autism, I am not an expert on technical aspects of ECoG or animal models of ASD or marmoset studies. My review therefore should be complemented by others.

The hypothesis is that there is diversity in predictive coding across cortical hierarchies in ASD, explaining heterogeneous results. I agree this is a likely explanation, and like the way of testing it here, with the local global oddball, and using a quantitative model. The results suggest that indeed there are differences across the hierarchy and between VPA-treated animals, whereas the three non-VPA animals are more similar to each other and have more close-to-optimal predictive coding processing across the hierarchy. The results advance our understanding of ASD (as seen through this animal model) in terms of the influential predictive coding theory. In particular, it provides evidence that will move us away from overly simplistic theories, and it provides a simple computational model useful for asking questions about a hierarchical task (important to contrast differences in processing across different stimulus complexity levels). The computational model has been previously used in macaques and in humans, using both ECoG and EEG, and is therefore accumulating evidence for its applicability.

Overall, from my perspective, this is a good study, with a high degree of sophistication, which advances our understanding of both predictive coding and ASD. As such, I would like to see it published.

I wonder if there should be more of a distinction in the set-up, and discussion, between, on the one hand, predictive coding accounts that contrast top-down vs bottom-up precisions, and, on the other hand, those that focus on beliefs about volatility (such as Lawson). The variability in the

results may fit better with problems with beliefs about volatility than with more basic predictive coding accounts focusing primarily on weak priors or strong prediction errors. The Discussion touches upon these types of explanations, focusing on precision weighting and context dependence; it might be worth making the link to volatility beliefs here? The authors mention the HGF which indeed would be useful in modelling volatility and could address what is going on in this ASD model, but they mention that it is not suitable for a hierarchical setting such as local-global MMN tasks. I did not quite understand this, see for example these two studies, which seem in the ballpark:

Daniel J. Hauke, Colleen E. Charlton, André Schmidt, John D. Griffiths, Scott W. Woods, Judith M. Ford, Vinod H. Srihari, Volker Roth, Andreea O. Diaconescu, Daniel H. Mathalon, Aberrant Hierarchical Prediction Errors Are Associated With Transition to Psychosis: A Computational Single-Trial Analysis of the Mismatch Negativity, *Biological Psychiatry: Cognitive Neuroscience and Neuroimaging*, Volume 8, Issue 12, 2023, Pages 1176-1185, <https://doi.org/10.1016/j.bpsc.2023.07.011>.

Marshall L, Mathys C, Ruge D, de Berker AO, Dayan P, Stephan KE, et al. (2016) Pharmacological Fingerprints of Contextual Uncertainty. *PLoS Biol* 14(11): e1002575.

<https://doi.org/10.1371/journal.pbio.1002575>

I am not suggesting the authors conduct a HGF model, but it is important to not unduly dismiss HGF, especially because it would likely be an advance in understanding. The field is fastmoving, and this new generalised HGF might well be applicable <https://arxiv.org/abs/2305.10937>

It is true that the findings here could help move towards a biomarker for ASD, but I'd hedge this claim somewhat more than 'potential' and perhaps be careful with "our study lays the groundwork for identifying a comprehensive, multi-tiered, and mechanistic neural marker for ASD". The field is littered with failed suggested biomarkers after all, and this study has two animals showing different patterns.

Reviewer #1 (Remarks to the Author):

Article revision

Title

Erroneous Predictive Coding Across Brain Hierarchies in a Non-Human Primate Model of Autism Spectrum Disorder

Overview

Atypical sensory experience in Autism Spectrum Disorder (ASD) is associated with irregular predictive processing across brain hierarchies. This study employed high-density electrocorticography to record neural activity during passive listening to a local-global oddball paradigm. The subjects were typical developing and valproic acid-induced female and male marmosets.

The results reveal the diverse configuration of prior beliefs, demonstrating the coexistence of overly precise and weak priors within the valproic acid-induced group, and a common hypersensitivity. These findings support the existence of heterogeneous sensory profiles within the ASD population.

We appreciate the reviewer's understanding of the key points of our paper.

Major comments

Introduction

- Page 3, lines 2-6: The introductory paragraph is quite lax and somewhat weak. More importantly, it is not informative on the environmental valproic acid-induced animals used in this study. We recommend addressing the validity and robustness of autistic traits shown by this model, and its wide use in basic research.

We agree with the reviewer's suggestions. In the revised manuscript, we maintain the first sentence for a general overview of ASD, and then immediately address the significance of sensory atypicalities in ASD research:

“Autism spectrum disorder (ASD) is a neurodevelopmental condition that includes challenges in social interaction and communication, repetitive behaviors, sensory hypo/hypersensitivity, and difficulties adapting to change. A leading mechanistic investigation of ASD focuses on its atypical sensory perception, such as hypersensitivities to light or sound, which is reported in around 90% of autistic adults ¹.” (page 3, lines 2-6)

We address the validity and robustness of the VPA animal model in the revised Introduction, including 4 additional references (37-40) on the marmoset model:

“To test this hypothesis, we extract prediction-error signals across hierarchies and examine their atypical characteristics using a marmoset model of ASD ³⁶. This model was created by administering valproic acid (VPA) during pregnancy, a well-known risk factor for ASD. Maternal exposure to VPA induces ASD-like behavioral abnormalities and stress responses in marmoset offspring ^{37,38}. Importantly, the transcriptomic profile of the cerebral cortex in VPA-treated marmosets—reflecting the interactions between genetic and environmental factors—shows strong correlations with post-mortem brain transcriptomes from human ASD populations ³⁶. This correlation has not been observed in any rodent models previously used. Furthermore, the observed similarity in dysregulated neuronal gene networks between VPA-treated marmosets and humans with ASD suggests that this animal

model could accurately represent major ASD subtypes, whose existence has been proposed due to weak interactions within individual gene networks^{39,40}.” (page 4 line 24 to page 5 line 11)

- Page 3, line 10: *Sensory atypicality in ASD is explained by different theories such as “intense world theory”, “temporal binding hypothesis”, “atypical hierarchical information processing” (Balasco et al., 2019). It should be clarified that failures in Bayesian inference are one theory among others attempting to explain atypical sensory experiences in ASD.*

We agree with the reviewer’s suggestions. In the revised manuscript, we have incorporated the theories proposed by the reviewer, along with other theories including the “enhanced perceptual functioning theory” and the “weak central coherence theory.” These are addressed in the first paragraph of the Introduction:

“Several theoretical models have been proposed to explain these sensory atypicalities. The enhanced perceptual functioning theory² and the weak central coherence theory³ suggest that individuals with ASD have a bias toward locally-oriented processing, attending to details rather than global patterns. The temporal binding theory⁴ suggests that individuals with ASD integrate sensory information over a prolonged time window, leading to a blurred or smeared perception of stimuli. The intense world theory⁵ posits that excessive functioning of neural circuits causes heightened low-level sensory perception in ASD, leading to an overwhelming and fragmented sensory experience of the world. While these frameworks significantly shape our understanding of ASD, they do not directly correspond to the underlying neural mechanisms.” (page 3, lines 6-15)

- Page 3 and 4, lines 25 and 1-6, respectively: *“Experimentally, prediction-error signals have been probed by surprise responses when expected stimuli are replaced or omitted” is a lax introduction to surprise responses. We recommend rephrasing to emphasize that mismatch negativity (MMN), the brain response to the violation of the regularity, is considered the neural substrate of prediction error (Fitzgerald and Todd, 2020). The lines 1-6 are confusing. The MMN is not one example, but the brain response to the surprise event interrupting a regular context. This response is recorded by EEG, MEG, and fMRI, as some examples (Garrido et al., 2009). The authors must rephrase this paragraph for the sake of clarity.*

We agree that the MMN is considered a neural substrate of prediction error. However, not all prediction errors are represented by MMN. For instance, the P3b, a subcomponent of the P300, is thought to represent prediction error at a higher hierarchical level (e.g., Wacongne et al., 2011), and the N400 is associated with semantic prediction error (e.g., Hodapp and Rabovsky, 2021).

Furthermore, MMN is defined as a contrast response in event-related potentials, applicable to both EEG and MEG. Hence, mentioning only EEG in our original manuscript was indeed incomplete. In Garrido et al., 2009, simultaneous fMRI-EEG recording was utilized, with fMRI aiding in the interpretation of the MMN detected in EEG. However, we have deemed this detail non-essential for our argument and omitted it in the revised manuscript. In the revision, we also introduce an argument that the modulation of MMN across hierarchies may be disrupted in ASD, supported by two new references (30 and 35), which provides clearer motivation for our hypothesis:

“Experimentally, prediction-error signals have been probed by surprise responses when expected stimuli are replaced or omitted. A key neural indicator of prediction error is the mismatch negativity (MMN), an event-related potential triggered by unexpected oddball stimuli, has been shown to vary in amplitude between individuals with ASD and typically developing individuals^{29–31}. However, meta-analyses on these reports revealed no consistent trend in these differences^{32,33}. Furthermore, the MMN amplitude can be influenced by statistical regularities over longer timescales^{34,35}, with this modulation found to be reduced in ASD³⁰. This suggests that the interaction of prediction errors across hierarchical levels may be disrupted in ASD.” (page 4, lines 11-19)

- Page 4, lines 8-11: Single- or multiple neural representations of predictive coding computations are provided by the complexity of the oddball paradigm (i.e., classical oddball paradigm vs. local-global oddball paradigm, respectively). The MMN is elicited by violating the regularity of either oddball paradigm. Indeed, the MMN frames the experimental design chosen for this study on local-global processing. The authors should reformulate this paragraph.

We also agree that MMN or MMN-liked responses should appear when transition of stimuli are learned and violated, regardless of the paradigm (e.g. oddball, local-global, paired stimuli disclination) or even domains (e.g. auditory, visual, somatosensory). To avoid confusion in the revision, we simply state that a single neural representation cannot capture the configuration of prediction errors across hierarchies:

“We hypothesize that the heterogeneous behavioral and neural evidence is caused by a diverse combination of erroneous predictive-coding computations occurring across cortical hierarchies, thus cannot be identified by a single neural representation where prediction-error signals across all hierarchies are mixed together.” (page 4, lines 21-24)

Methods

- Page 24, line 4: This study used female and male marmosets to address autistic traits without specifying biological sex of typical developing and valproic acid-induced animals. In the induced group, the diverse findings on prior beliefs may be influenced by sex, as sex differences are expected in the typical and atypical brain (Ruigrok et al., 2014). Moreover, ASD's diagnosis is biased by sex (Gu et al., 2023). Thus, it is crucial to specify whether sex is a factor of variability when analyzing neural responses in the valproic acid-induced group.

In the original manuscript, an error was made regarding the sex of the animals used in the study; they are all males. This has been corrected in the Methods section (page 27, line 4).

In the revised manuscript, we have also included a new section titled "Limitations and future work" in the Discussion. This section addresses the points raised by the reviewer and incorporates the references suggested:

“A major limitation of our study is the small number of VPA-treated marmosets, which was primarily due to the complexities involved in administering VPA and implanting hemisphere-wide ECoG. This restricts our ability to assess the diversity and spectrum of erroneous predictive coding and to identify potential ASD subtypes. Additionally, as the diagnosis of ASD can be influenced by sex⁸⁸, and sex differences are anticipated in both

neurotypical and atypical brain development⁸⁹, the fact that our study groups comprised only male animals precludes an investigation into whether sex contributes to the observed variability in erroneous predictive coding.” (page 25, lines 11-18)

- Page 25, line 8: *The array was implanted into the right or left hemisphere of the marmosets. The typical developing group had 2 right and 1 left implantations, and the valproic acid-induced group had 2 left implantations. Apparently, hemisphere location does not influence neural recordings. The authors should record a single hemisphere per experimental design to avoid future biases (i.e., Devlin et al., 2003). Describe as a limitation to avoid in future studies.*

In the newly added "Limitations and future work" section of the Discussion, we address the limitation pointed out by the reviewer and incorporate two additional references (90-91), including one recommended by the reviewer:

“Moreover, in our study, the unexposed group underwent both right and left implantations, while the VPA-treated group only had left implantations. Including more subjects with implantations on each side will allow us to investigate potential atypical lateralization in ASD⁹⁰ and reduce biases associated with functional lateralization⁹¹” (page 25, lines 18-21)

Minor comments

Introduction

- Page 4, line 8-9: *Correct the verb in the following sentence: “We hypothesize that the heterogeneous behavioral and neural evidence is caused by a diverse combination of erroneous predictive-coding computations ”occurring” across cortical hierarchies [...]”*

We correct it in the revision (page 4, line 22).

Reviewer #2 (Remarks to the Author):

I want to state first that my field of expertise is limited to theoretical and empirical discussions around predictive coding and autism, I am not an expert on technical aspects of ECoG or animal models of ASD or marmoset studies. My review therefore should be complemented by others.

The hypothesis is that there is diversity in predictive coding across cortical hierarchies in ASD, explaining heterogeneous results. I agree this is a likely explanation, and like the way of testing it here, with the local global oddball, and using a quantitative model. The results suggest that indeed there are differences across the hierarchy and between VPA-treated animals, whereas the three non-VPA animals are more similar to each other and have more close-to-optimal predictive coding processing across the hierarchy. The results advance our understanding of ASD (as seen through this animal model) in terms of the influential predictive coding theory. In particular, it provides evidence that will move us away from overly simplistic theories, and it provides a simple computational model useful for asking questions about a hierarchical task (important to contrast differences in processing across different stimulus complexity levels). The computational model has been previously used in macaques and in humans, using both ECoG and EEG, and is therefore accumulating evidence for its applicability.

Overall, from my perspective, this is a good study, with a high degree of sophistication, which advances our understanding of both predictive coding and ASD. As such, I would like to see it published.

We appreciate the reviewer's understanding of the key points of our paper.

I wonder if there should be more of a distinction in the set-up, and discussion, between, on the one hand, predictive coding accounts that contrast top-down vs bottom-up precisions, and, on the other hand, those that focus on beliefs about volatility (such as Lawson). The variability in the results may fit better with problems with beliefs about volatility than with more basic predictive coding accounts focusing primarily on weak priors or strong prediction errors.

In the initial version, we classified the compromised volatility described by Lawson et al. (2017) under the category of irregular prediction updates. Following the reviewer's suggestion, we acknowledge that this differs from the slow prediction update identified by Lieder et al. (2019), and should be a distinct category within Bayesian frameworks. Consequently, we have revised the Introduction to reflect this distinction:

“Through the Bayesian lens, sensory atypicalities in ASD could arise from various factors: overly precise sensory observations¹⁰⁻¹², weak prior beliefs^{7,13}, slow updates of these beliefs¹⁴, and imbalanced control of precision^{13,15,16}, and overestimation of environmental volatility¹⁷.” (page 3, lines 18-21)

To date, only overestimation of volatility has been documented in ASD, which typically results in reduced surprise. This does not adequately explain our findings of increased prediction errors, as observed in our studies with monkey Ca. We discuss this issue in the revised Discussion, incorporating two additional references (70-71):

“Beyond irregular predictions, another consideration is the potential misestimation of the volatility or uncertainty associated with these predictions. This “belief in volatility” is thought to play a crucial role in driving learning⁷⁰, and in modulating the balance between exploiting existing knowledge and exploring new possibilities⁷¹. Studies have shown that adults with ASD tend to overestimate the volatility of sensory environments, leading to a greater tendency to ignore unexpected stimuli and experience less surprise¹⁷. However, it is still uncertain whether underestimations of volatility can occur in ASD, possibly leading to contrasting patterns of prediction errors. To explore this, further manipulation of environmental uncertainty and a detailed examination of the learning processes are required.” (page 23, lines 6-15)

The Discussion touches upon these types of explanations, focusing on precision weighting and context dependence; it might be worth making the link to volatility beliefs here? The authors mention the HGF which indeed would be useful in modelling volatility and could address what is going on in this ASD model, but they mention that it is not suitable for a hierarchical setting such as local-global MMN tasks. I did not quite understand this, see for example these two studies, which seem in the ballpark:

Daniel J. Hauke, Colleen E. Charlton, André Schmidt, John D. Griffiths, Scott W. Woods, Judith M. Ford, Vinod H. Srihari, Volker Roth, Andreea O. Diaconescu, Daniel H. Mathalon, Aberrant Hierarchical Prediction Errors Are Associated With Transition to Psychosis: A Computational Single-Trial Analysis of the Mismatch Negativity, Biological Psychiatry: Cognitive Neuroscience and Neuroimaging, Volume 8, Issue 12, 2023, Pages 1176-1185, <https://doi.org/10.1016/j.bpsc.2023.07.011>.

Marshall L, Mathys C, Ruge D, de Berker AO, Dayan P, Stephan KE, et al. (2016) Pharmacological Fingerprints of Contextual Uncertainty. PLoS Biol 14(11): e1002575. <https://doi.org/10.1371/journal.pbio.1002575>

We thank the reviewer for the suggestion and have included both references in the revised manuscript. However, it is important to note a clear distinction in the hierarchical representation between predictive coding and HGF. We address this difference in the revised Discussion section:

“A Bayesian model known as the hierarchical Gaussian filtering (HGF) has emerged as a promising candidate for understanding prediction updating and error minimization⁷². This model employs precision-weighted prediction errors^{23,25,73} and has been utilized to investigate prediction-error signals in the brain during learning⁷⁴⁻⁸⁰. HGF models the stimulus probability (first level), the tendency of the change of the probability over a longer timescale (second level), the volatility of the tendency (third level), where higher levels determine the rate of change at lower levels. It’s important to note that this hierarchical representation differs from predictive coding, where higher levels influence the state of lower levels. Therefore, the sequence structure in the local-global paradigm is not simply a tendency of stimulus probability or its volatility. Nevertheless, exploring whether the higher levels of HGF can capture sequence probability will be a worthwhile endeavor.” (page 23 line 21 to page 24 line 6)

We also discuss this in the future work section in Discussion, including a new generalized HGF (reference 92):

“Another future plan is to examine the single-trial learning using a more generalized HGF that can bridge more directly to predictive coding⁹²” (page 26, lines 3-4)

It is true that the findings here could help move towards a biomarker for ASD, but I'd hedge this claim somewhat more than 'potential' and perhaps be careful with "our study lays the groundwork for identifying a comprehensive, multi-tiered, and mechanistic neural marker for ASD". The field is littered with failed suggested biomarkers after all, and this study has two animals showing different patterns.

We agree with the reviewer and acknowledge our oversight. Accordingly, we have moderated our claims throughout the revised manuscript. In Discussion:

“By linking computational theories with their neural underpinnings, our study contributes to the foundation for potentially identifying a comprehensive, multi-tiered, and mechanistic neural marker for ASD.” (page 20, lines 7-10)

We have also added the term "potential" to the Abstract to better qualify our claims:

“Our results demonstrate the coexistence of the two primary Bayesian accounts of ASD: overly-precise sensory observations and weak prior beliefs, and offer a potential multi-layered biomarker for ASD, which could enhance our understanding of its diverse symptoms.” (page 2, lines 14-16)

REVIEWERS' COMMENTS:

Reviewer #1 (Remarks to the Author):

The author has fully addressed my concerns.

Reviewer #2 (Remarks to the Author):

Thank you for these revisions, which sufficiently addresses my comments.